# Mechanical and Hydraulic Properties of Recycled Concrete Aggregates Mixed with Clay Brick Aggregates and Particle Breakage Characteristics for Unbound Road Base and Subbase Materials in Vietnam

Trong Lam Nguyen [1], Van Tuan Nguyen [1,*], Hoang Giang Nguyen [2,3], Akihiro Matsuno [4], Hirofumi Sakanakura [5] and Ken Kawamoto [3,4]

1 Department of Building Materials, Hanoi University of Civil Engineering, No. 55 Giai Phong Street, Hai Ba Trung District, Hanoi 11616, Vietnam; lamnt@huce.edu.vn
2 Department of Civil and Industrial Construction, Hanoi University of Civil Engineering, No. 55 Giai Phong Street, Hai Ba Trung District, Hanoi 11616, Vietnam; giangnh@huce.edu.vn
3 Innovative Solid Waste Solutions (Waso), Hanoi University of Civil Engineering, No. 55 Giai Phong Street, Hai Ba Trung District, Hanoi 11616, Vietnam; kawamoto@mail.saitama-u.ac.jp
4 Graduate School of Science and Engineering, Saitama University, 255 Shimo-okubo, Sakura-ku, Saitama 3388570, Japan; matsuno2017@mail.saitama-u.ac.jp
5 Material Cycles Division, National Institute for Environmental Studies, 16-2 Onogawa, Tsukuba City 3058506, Japan; sakanakura@nies.go.jp
* Correspondence: tuannv@huce.edu.vn; Tel.: +84-90-988-6386

**Abstract:** The construction industry is one of the key industries with high potential for the circular economy; the promotion of reuse and recycling of construction and demolition waste (CDW) is essential for sustainable urban development. In this study, a series of compaction, California bearing ratio, saturated hydraulic conductivity, and particle breakage tests of well– and poor–graded mixtures of recycled clay brick aggregates (RCBs) and recycled concrete aggregates (RCAs) with maximum aggregate diameters of 19, 25, and 37.5 mm were carried out to examine the practical application of those mixtures to unbound roadbed materials in Vietnam. The experimental results suggest that the maximum amount of RCBs added to RCAs should be less than 30% when applied to unbound roadbed materials. In addition, it was found that the mixing proportions of RCBs and RCAs and the maximum aggregate diameter, gradation of aggregates, and initial moisture condition control the saturated hydraulic conductivity. Further, the particle breakage characteristics under compaction were carefully examined, and it was found that the percentage increment/decrement, as well as a newly introduced method of estimating the mixing proportions of RCAs and RCBs in the fine fraction (<2.36 mm), is effective in understanding the mechanism of particle breakage of RCA and RCB mixtures.

**Keywords:** recycled concrete aggregate; recycled clay brick aggregate; compaction; road base and subbase; California bearing ratio (CBR); hydraulic conductivity; particle breakage

## 1. Introduction

Rapid urbanization and population growth cause adverse effects, such as unsustainable use of natural resources, environmental pollution, and degradation over large areas [1–3]. In particular, the construction industry not only consumes enormous natural resources, but also generates a large amount of construction and demolition waste (CDW) in the process of urban renewal and redevelopment [2,4,5]. The construction industry, on the other hand, is considered one of the key industries that has a high potential for the adoption of a circular economy [6–9]. Asif et al. [6], for example, indicated that the construction industry is responsible for the consumption of 40% of the total energy and

natural resources consumed by the global economy. Reuse and recycling of CDW, as well as the minimization of waste generation from the construction industry, therefore, fit well into the promotion of recycling economy practices [10–13].

In fact, CDW rich in recyclable concrete and clay brick has a strong potential for reuse and recycling due to the high utility of recycled materials (mostly in the field of civil engineering) and the large market, such as in the use of recycled aggregates in road construction [14–17]. In addition, the promotion of reuse and recycling of CDW reduces environmental impacts, such as landfill and resource depletion, air and water pollution, and high energy consumption [2,18,19]. Recently, many countries have begun to promote the reuse and recycling of CDW, and the percentages of reuse and recycling have reached 97% in Japan, 90% in the Netherlands and UK, 81% in Denmark, 70% in the USA, 59% in France, and 85% in Germany [20–23]. In many developing countries, including Vietnam, however, the reuse and recycling of CDW are still low and are generally <10% [24–26].

Among developing countries, Vietnam has been facing rapid urbanization and population growth due to economic growth on all fronts, and it has had especially remarkable growth in the construction industry [2,4,8]. Many construction and demolition activities, including renovation and demolition of old buildings and structures, are being conducted all over the country, especially in big cities such as Hanoi, Ho Chi Minh, Haiphong, and Da Nang. These activities have generated a huge amount of CDW; e.g., approximately 2000 apartment buildings (about 90% of the total) built in 1970–1980 are now seriously degraded and require renovation and renewal. The Vietnamese government issued Resolution 34/2007/NQ-CP [27] in 2007 on measures to renovate the degraded apartment blocks by 2015. Currently, the total CDW generated has reached approximately 1.9 million tons per year, or 10–12% of total solid waste in Vietnam. Only 1–2% of CDW generated, however, has been reused and recycled, and the remaining waste is mainly dumped improperly on-site and/or off-site in landfills [24]. Therefore, it has become an urgent issue to establish a proper CDW management system and to promote the reuse and recycling of CDW in order to reduce the consumption of natural resources and for prevent improper dumping in Vietnam [2,4,8].

Among the CDW generated from construction projects and old building demolitions, concrete and clay brick waste is commonly recycled and applied for many civil engineering purposes, such as in road base materials, recycled aggregates for concrete, and backfilling materials [28,29]. One of the especially promising applications for recycled concrete is use as unbound base and subbase materials in road construction due to its easy applicability to road base and subbase materials, resulting in a high contribution to the increase in CDW recycling [14,15,17]. Many studies have been addressed the mechanical properties of RCAs to improve the performance of unbound road base and subbase materials. For example, Thai et al. [30] conducted an in-depth review and assessment of the effects of the size and type of aggregates on the mechanical properties of RCAs. They indicated that the mechanical indices depend significantly on the type of aggregate (e.g., RCAs mixed with recycled clay brick aggregates), on the maximum size of the aggregates ($D_{max}$), and their gradation (i.e., grain size distribution and fine content). Moreover, many studies have targeted the mechanical properties of recycled concrete aggregates (RCAs) mixed with recycled clay brick aggregates (RCBs) to evaluate the applicability to unbound road base and subbase materials. Poon and Chan [14] revealed that the use of RCBs to replace RCAs reduced the maximum dry density (MDD) and CBR values and increased the optimum moisture content (OMC) of the subbase materials compared with those of natural subbase materials. The authors also indicated that a minimum soaked CBR value of 35 can meet the requirements for producing a subbase in Hong Kong with a blend RCAs and RCBs. Aatheesan et al. [15,16,31,32] studied the potential application of the combination of RCB and RCA/crushed rock (basalt) for pavement subbase and drainage systems with experimental programs, including hydraulic conductivity, mechanical properties such as CBR, Los Angeles abrasion loss, consolidated drained static triaxial, and repeated loading triaxial tests. The experimental results indicated that about 25–30% RCB could be safely

added to Class 3 RCA/crushed rock blends for pavement subbase applications of the Roads Corporation of Victoria state in Australia. However, this value was limited to 15% considering the degree of breakdown after compaction occurring in the RCB blend. Similar findings were reported by Cameron et al. [17].

As described in the previous paragraphs, many studies have been performed especially to examine the mechanical properties of RCB (and/or RCM) and RCA mixtures for application to road base and subbase materials. However, limited studies have investigated the particle breakage of compacted samples, especially to characterize the fraction of concrete and clay brick retained in each fraction of aggregate size and fineness (typically, <2 mm) after compaction, even though the mixed samples were composed of two aggregates (RCA and RCB) with different physico-chemical and mechanical properties, such as element composition, water absorption, hardness, and resilience. This study, therefore, especially targeted the practical application of RCAs and RCBs for roadbed materials in Vietnam, and the objectives were (i) to investigate the effects of $D_{max}$ and gradation of RC mixed with RCBs on the mechanical and hydraulic properties, (ii) to examine the particle breakage characteristics for compacted RCAs mixed with RCBs at different initial moisture contents based on the percent increment/decrement of each particle fraction before and after compaction, and (iii) to suggest a useful method for estimating the percentage of concrete and clay brick in fines.

## 2. Materials and Methods

### 2.1. Materials

Concrete and clay brick waste was collected from the Thanh Tri CDW dumping site in Hanoi, Vietnam. The waste samples were first crushed with a jaw and hammer mill crusher to adjust the particle (aggregate) size to less than 50 mm. Next, the crushed samples of RCA and RCB were mixed at different proportions to become four types of samples with different values $D_{max}$ and gradation, and the samples were labeled $D_{max}$ = 19 mm, $D_{max}$ = 25 mm, $D_{max}$ = 37.5 mm (well–graded), and $D_{max}$ = 37.5 mm (poor–graded). The mixing proportion of RCB to RCA (f, in percent on the basis of dry mass) was set to 0%, 10%, 30%, 50%, 70%, or 100% for the tested samples, with three $D_{max}$ of 19, 25, and 37.5 mm (well–graded). For the samples with $D_{max}$ of 37.5 mm (poor–graded), the f values were set to 0%, 20%, 40%, 60%, or 100%. The samples of mixed RCA and RCB were named "RCA 100%", "RCA 50% + RCB 50%", and "RCB 100%" in this study depending on the f values of the tested samples (see Table 2). Photos of graded samples before mixing and compaction are exemplified in Figure 1.

The basic physical and chemical properties of the RCA and RCB were determined by complying with some standards, such as the American Standards for Testing of Materials (ASTM), Japan Industrial Standards (JIS), the standards of the Japanese Geotechnical Society (JGS), and Vietnamese standards, and they are summarized in Table 1. The measured physical and chemical properties of the RCA were in the range of general values reported in previous studies [33–35] and met the technical requirements for road base and subbase materials regulated by Vietnamese standards TCVN 8859:2011 [36] and TCVN 8857:2011 [37], as well as those of the Japan Road Association (JRA, 2010) [38]. The $w_{abs}$ values and the LA values of the RCB were higher than those of the RCA, indicating that the RCB was more abrasive and adsorbed more water than the RCA. The main differences in the chemical compositions of the RCA and RCB are shown in terms of the percentages of $SiO_2$, $CaO$, and $Al_2O_3$, and this can be understood because the RCA originated from cementitious materials rich in Ca and the RCB was made from soils rich in Si and Al. The particle size distributions (PSDs) of the four types of tested samples are shown in Figure 2. The PSDs of samples (before compaction in the figure) were adjusted to the gradation ranges given in the Vietnamese standards of crushed stone and natural aggregates for road base and subbase materials in this study [36,37]. For reference, the standard gradation, adaptations, and technical specifications of aggregates for unbound road base and subbase materials in TCVN 8859 [36] and TCVN 8857 [37], as well as JRA [38], are summarized in Appendix A.

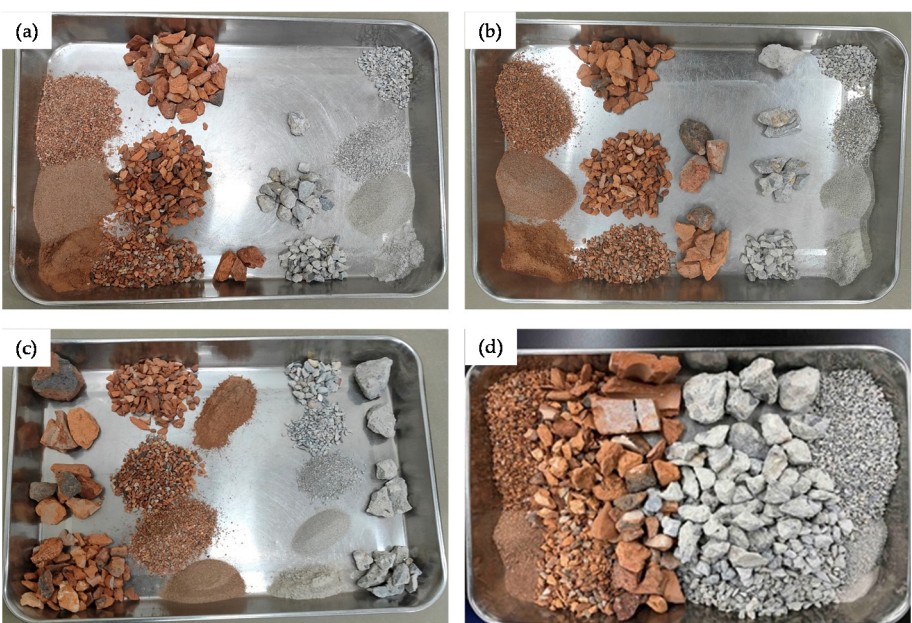

**Figure 1.** Sieved and air-dried samples before mixing ($w_i$ = 0.5~3%). (**a**) $D_{max}$ = 19 mm(RCA 50% + RCB 50%), (**b**) $D_{max}$ = 25 mm (RCA 50% + RCB 50%), (**c**) $D_{max}$ = 37.5 mm (well–graded; RCA 50% + RCB 50%), and (**d**) $D_{max}$ = 37.5 mm (well–graded; RCA 60% + RCB 40%).

**Table 1.** Basic physical and chemical properties of the RCA and RCB in this study.

| Samples | $G_s$ | $w_{AD}$ (%) | $w_{abs}$ (%) <4.75 mm | $w_{abs}$ (%) ≥4.75 mm | pH [1] | EC [1] (mS/cm) | LA (%) | MgO | Al$_2$O$_3$ | SiO$_2$ | CaO | Fe$_2$O$_3$ | Others |
|---|---|---|---|---|---|---|---|---|---|---|---|---|---|
| RCA | 2.72 | 0.8 | 8.5 | 5.2 | 11.2 | 4.8 | 34 | 7.3 | 6.3 | 35.4 | 30.9 | 1.7 | 18.4 |
| RCB | 2.64 | 0.3 | 14 | 13 | 10.7 | 0.0 | 46 | 0.9 | 17.9 | 68.2 | 0.6 | 7.5 | 4.9 |

$G_s$: Specific gravity, $w_{AD}$: air-dried water content, $w_{abs}$: water absorption, EC: electrical conductivity, LA: Los Angeles abrasion. [1] pH was measured by using a 1 mol KCl solution (S:L = 1:2.5), and EC was measured by using distilled water (S:L = 1:5) for sieved samples <2 mm. [2] The chemical component was measured with a fundamental parameter method of energy-dispersion X-ray spectrometry (FP-EDX) for sieved samples <2 mm.

## 2.2. Compaction, CBR, and Saturated Hydraulic Conductivity Tests

In this study, to maintain the performance and longevity of the road pavement structure, some mechanical properties of the RCA, such as the compaction, California bearing ratio (CBR), saturated hydraulic conductivity, and particle breakage, were analyzed after the compaction tests were conducted for application as unbound road base and subbase materials.

### 2.2.1. Compaction Test

The compaction test was performed by following the modified Proctor compaction method described in TCVN 12790 [39] and ASTM D 1557 [40]. A cylindrical mold with a size of (d × h) 150 × 125 mm was used to compact the tested samples with a 4.54 kg rammer dropped at a height of 457 mm. The samples were compacted with five layers and 56 blows (compaction energy of 2631 kJ/m$^3$). To determine the relationship between the initial moisture content ($w_i$, kg/kg in %) and measured dry density (g/cm$^3$) of the tested samples, the initial moisture contents of tested samples were adjusted, ranging from air-dried (~0.5–3%) to ~15%, by either air-drying or adding water to the samples from the field moisture content of ~5–9%.

### 2.2.2. CBR Test

The CBR test was performed using the procedure outlined in TCVN 12792 [41] and AASHTO T193 [42]. Samples were compacted in a (d × h) 150 × 125 mm cylindrical mold using a 4.54 kg rammer dropped at a height of 457 mm. The initial $w$ of the tested samples was adjusted to about 8% (approximately corresponding to the maximum dry density in the compaction curve; see Figure 3). The tested samples were compacted in three layers by applying 10, 30, and 65 blows per layer. The compacted samples were immersed in water for 96 h, and their deformation was recorded by using a dial gauge. At the end of the soaking period, the free water was collected. After that, a load was applied with a standard 50 mm diameter plunger into the sample at the rate of 1.0 mm/min. Reading of the load was taken at penetrations of less than 12.5 mm depth. In this study, the CBR values of the tested samples at two different degrees of compaction (K) of 95% and 98% were calculated for penetrations of 2.5 and 5.0 mm (CBR2.5 and CBR5).

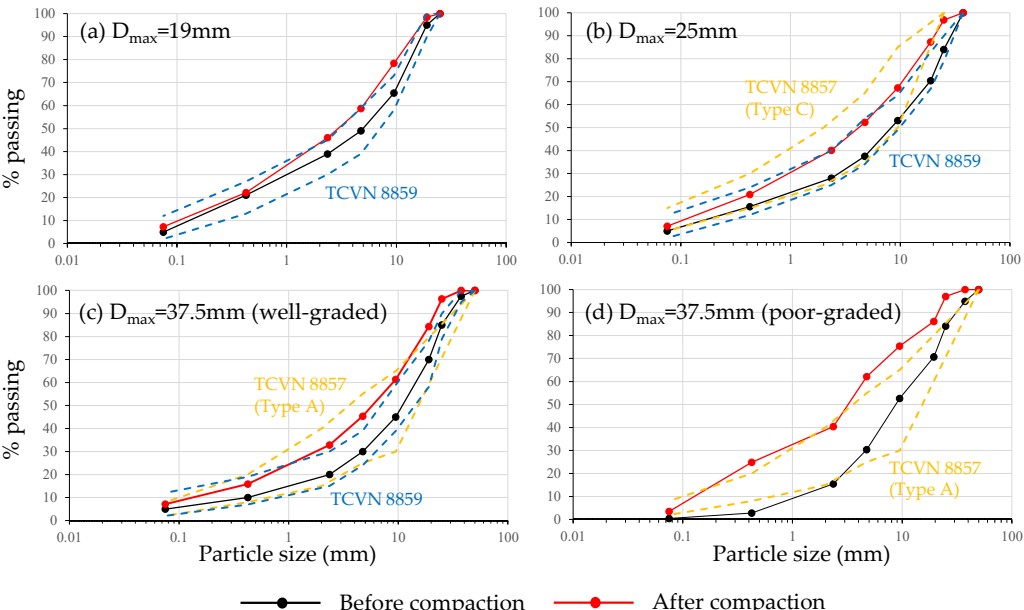

**Figure 2.** Particle size distributions (PSDs) for the tested samples before and after compaction ($w_i$ = ~8%: MDD). (**a**) $D_{max}$ = 19 mm, (**b**) $D_{max}$ = 25 mm, (**c**) $D_{max}$ = 37.5 mm (well–graded), and (**d**) $D_{max}$ = 37. 5mm (poor–graded). Upper and lower boundaries of PSDs for road base and materials prescribed in TCVN 8857 [37] and TCVN 8859 [36] are also given.

### 2.2.3. Saturated Hydraulic Conductivity Tests

The saturated hydraulic conductivities ($K_s$) of the compacted samples were measured by following ASTM D 5856 [43] and TCVN 12662 [44]. The tested samples were first kept in a bag at a constant temperature of 20 °C for more than 24 h to adjust $w_i$ to ~8%. Then, the samples were compacted in a cylindrical mold with a diameter of 150 mm and height of 125 mm with the modified Proctor compaction method. The compacted samples were next immersed in a water tank for more than 24 h to become fully saturated, and then the $K_s$ values of samples were measured with either the constant head (typically, $K_s$ > $10^{-3}$ cm/s) or falling head method (typically, $K_s$ < $10^{-4}$ cm/s).

### 2.3. Particle Breakage Analysis after the Compaction Test

In this study, the particle breakage index ($B_g$) of the tested samples was determined by following Marsal's method [45]. In this method, the $B_g$ value can be calculated based on the PSD curves before and after compaction and is given in Equation (1):

$$B_g = \sum_{i=1}^{n} \Delta p d_i, \tag{1}$$

where $\Delta pd_i$ is the positive difference in material retained on the $i$th sieve before and after the compaction test (% by weight). In addition, the retained mass in each size fraction was used to calculate the percent increment and/or decrement [35].

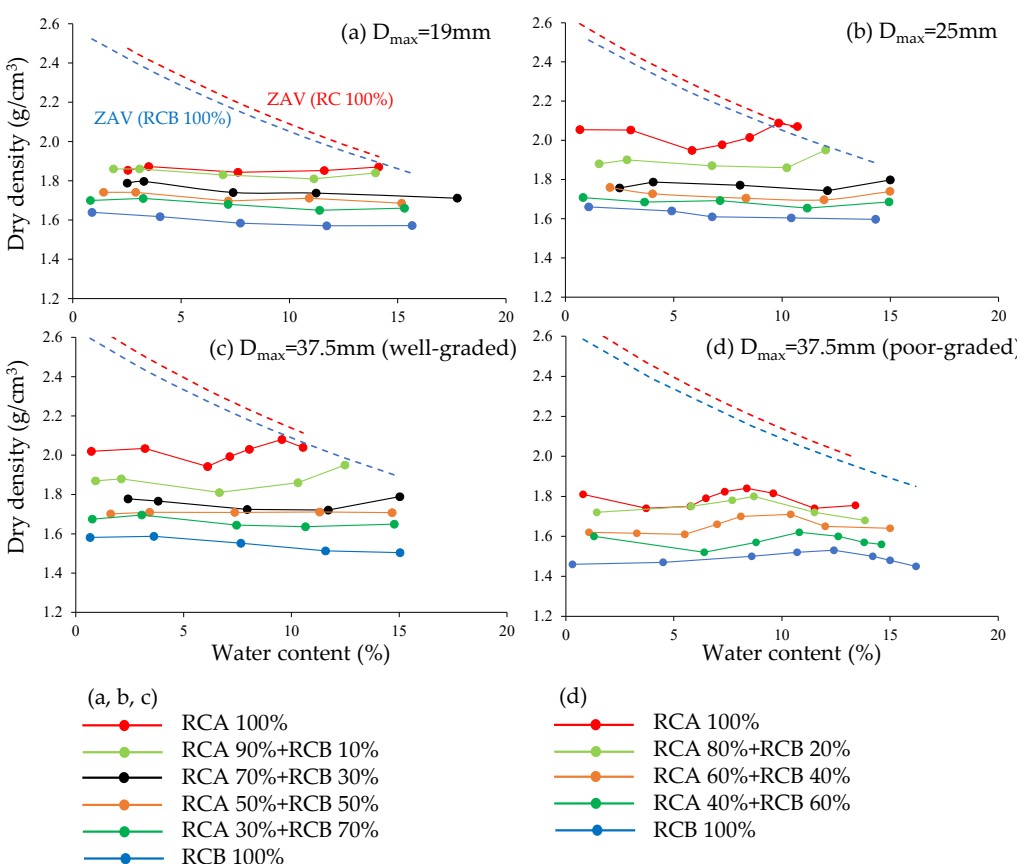

**Figure 3.** Compaction curves for the tested samples. (**a**) $D_{max}$ = 19 mm, (**b**) $D_{max}$ = 25 mm, (**c**) $D_{max}$ = 37.5 mm (well–graded), and (**d**) $D_{max}$ = 37.5 mm (poor–graded). Zero air void (ZAV) curves of RCA 100% and RCB 100% are also given.

## 3. Results and Discussion

### 3.1. Compaction Properties

The relationships between moisture content (i.e., $w_i$) and dry density (compaction curves) of the tested samples are illustrated in Figure 3, and the MDD values of the samples are summarized in Table 2. For all tested samples, the measured dry densities decreased with an increased mixing proportion of RCB to RCA ($f$, in %). Among the tested samples, the 100% RCA sample with $D_{max}$ = 25 mm (Figure 3b) and $D_{max}$ = 37.5 mm (well– graded) (Figure 3c) achieved high dry densities, and their MDD values exceeded 2.00 g/cm³. In the compaction curves, there was no clear peak in dry density through the adjusted $w_i$, indicating that no OMC existed. Except for the tested samples with $D_{max}$ = 37.5 mm (poor– graded) (Figure 3d), the MDD values of the 100% RCA sample were observed close to the zero void curves (ZVCs). Similarly to the tested results in this study, the unclear and/or absent peaks in dry density were also observed in previous studies using RCAs, RCBs, and recycled crushed glass [33,35]. Several studies, on the other hand, reported a clear peak (i.e., MDD) in the compaction curves (e.g., [46–48]). This evidence indicated that compacted dry densities and their dependence on $w_i$ were attributed highly to the differences in the maximum aggregate size, gradation, and aggregate materials. It was noted that the tested samples with $D_{max}$ = 37.5 mm (poor–graded) and with high $w_i$ close to the ZVC did not exist because free water was not retained in the sample, but was drained easily (Figure 3d). Comparing the tested compaction results from $D_{max}$ = 37.5 mm (well–graded) (Figure 3c)

and $D_{max}$ = 37.5 mm (poor–graded) shows that the gradation (i.e., PSDs) greatly affected the compaction for all samples of RCA, RCB, and their blends.

The MDD values measured for the tested samples were plotted against $f$ and are shown in Figure 4a. In the figure, data reported by previous studies on the compaction properties of RCAs mixed with RCBs [14,17,32,49] were also plotted. Overall, it can be seen that the MDD values in this study decreased gradually with increasing $f$, and a fitted function was given:

$$MDD = 2.0 \times 10^{-5} f^2 - 5.8 \times 10^{-3} f + 2.0 \ (r^2 = 0.82), \qquad (2)$$

**Table 2.** Measured MDD, $B_g$, CBR, and $K_s$ values for the tested samples. $B_g$ and CBR at K = 98% and 95%. $B_g$ and CBR values were measured in compacted samples at $w_i$ = ~8% (MDD).

| Samples | Mixing Proportion (%) | MDD (g/cm³) | $B_g$ | CBR at K = 98% (%) | CBR at K = 95% (%) |
|---|---|---|---|---|---|
| $D_{max}$ = 19 mm | RCA 100% | 1.87 | 8.3 | 132 | 81 |
| | RCA 90% + RCB 10% | 1.86 | 6.4 | 126 | 79 |
| | RCA 70% + RCB 30% | 1.80 | 12.0 | 146 | 78 |
| | RCA 50% + RCB 50% | 1.74 | 14.0 | 145 | 76 |
| | RCA 30% + RCB 70% | 1.71 | 10.5 | 126 | 79 |
| | RCB 100% | 1.64 | 14.8 | 185 | 117 |
| $D_{max}$ = 25 mm | RCA 100% | 2.03 | 7.2 | 299 | 249 |
| | RCA 90% + RCB 10% | 1.95 | 13.4 | 201 | 146 |
| | RCA 70% + RCB 30% | 1.80 | 16.9 | 123 | 79 |
| | RCA 50% + RCB 50% | 1.76 | 17.5 | 108 | 64 |
| | RCA 30% + RCB 70% | 1.71 | 18.1 | 139 | 71 |
| | RCB 100% | 1.66 | 27.1 | 198 | 132 |
| $D_{max}$ = 37.5 mm (well–graded) | RCA 100% | 2.06 | 9.3 | 291 | 248 |
| | RCA 90% + RCB 10% | 1.95 | 15.2 | 205 | 161 |
| | RCA 70% + RCB 30% | 1.79 | 19.4 | 120 | 84 |
| | RCA 50% + RCB 50% | 1.71 | 16.4 | 112 | 81 |
| | RCA 30% + RCB 70% | 1.70 | 18.7 | 156 | 109 |
| | RCB 100% | 1.59 | 21.8 | 117 | 76 |
| $D_{max}$ = 37.5 mm (poor–graded) | RCA 100% | 1.84 | 11.2 | 112 | 73 |
| | RCA 80% + RCB 20% | 1.80 | 12.2 | 117 | 79 |
| | RCA 60% + RCB 40% | 1.71 | 12.8 | 103 | 63 |
| | RCA 40% + RCB 60% | 1.60 | 16.4 | 55 | 36 |
| | RCB 100% | 1.53 | 21.0 | 50 | 37 |

The MDD values measured in this study were generally smaller than those reported in the literature, especially in the range of $f$ > 30%. This can mainly be attributed to the differences in the material properties of the RCA and RCB used, which are highly dependent on regional/national characteristics, including the materials for manufacturing concrete and clay bricks (sand, stones, cement, soils, etc.) and climate conditions, age of the source (i.e., age of the buildings demolished), and other factors (e.g., differences in crushing to produce aggregates). For reference, the data reported from other regions/countries (Hong Kong in Poon and Chan [14], Australia in Cameron et al. [17] and Arulrajah et al. [32], Egypt in Arisha et al. [49], and the USA in Diagne et al. [50]) are given in Figure 4a.

### 3.2. CBR Properties

The CBR values measured at K = 98% in the tested samples as a function of $f$ are shown in Figure 4b with data reported in the literature, and the CBR values measured at K = 98% and 95% in the samples are summarized in Table 2. For the tested samples with $D_{max}$ = 25 mm and $D_{max}$ = 37.5 mm (well–graded), the peak values of CBR at K = 98% measured were at RCA 100% ($f$ = 0%). Then, the values decreased with increasing $f$ and became local minima at $f$ = 30–50%. For the tested samples with $D_{max}$ = 19 mm, on the other hand, the measured CBR value at K = 98% became rather constant over all f values

and ranged between 126 and 146%, except for the value at RCB = 100% (*f* = 100%) of 185%. Among the samples tested in this study, the CBR values at K = 98% for $D_{max}$ = 37.5 mm (poor–graded) with *f* > 60% became lower than 100%, which corresponds to the technical requirements (subbase layer (A1) and base layer (A2 and B2) in TCVN 8859 [36] and for all adaptations except the base layer for A2 in TCVN 8857 [37], as shown in Table A1). This indicated that the maximum mixing of RCB to RCA was maintained at *f* ≤ 40% based on the technical requirements in Vietnam. Moreover, it is recommended that the maximum proportion of RCB to RCA should be maintained at *f* ≤ 30% for practical applications, such as in roadbed materials, in consideration of safety because of the inherently heterogeneous quality of RCBs generated from demolished sites.

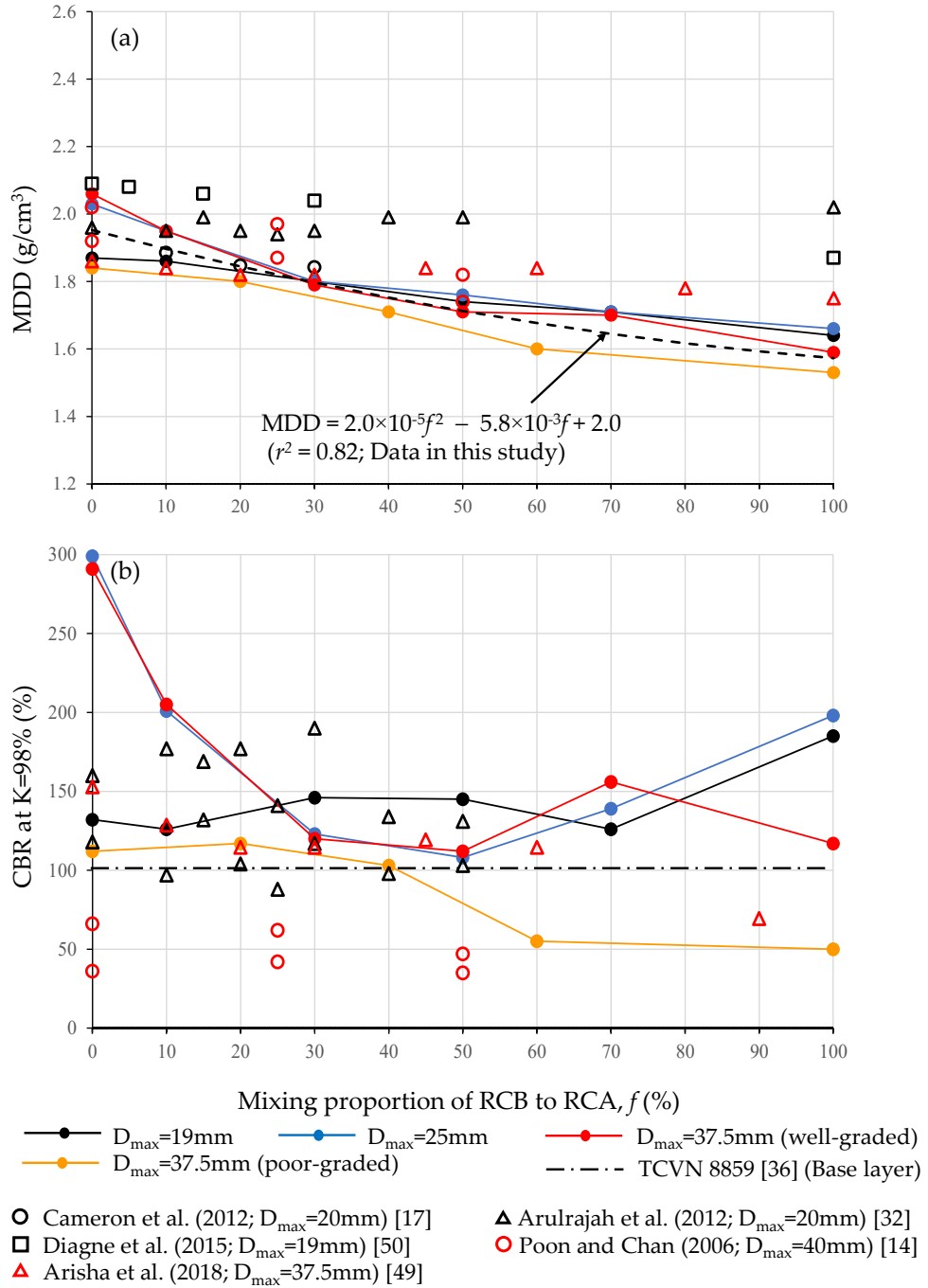

**Figure 4.** (**a**) MDD and (**b**) CBR at K = 98% as a function of mixing proportion (*f* in %) of RCB to RCA.

Comparing the CBR values measured at K = 98% to the values measured at 95% (the corrected CBR values) in Table 2, it can be seen that the CBR value at K = 98% was generally 1.5 times higher than the corrected CBR value. Among the corrected CBR values, the 100% RCB sample ($f$ = 100%) with $D_{max}$ = 37.5 mm (well graded) (corresponding to RM-40/base in Japan), the 40% RCA + 60% RCB sample ($f$ = 60%), and the 100% RCB sample ($f$ = 100%) of $D_{max}$ = 37.5 mm (poor–graded; corresponding to RC-40 (subbase) in Japan) did not satisfy the technical requirement for the base in JRA [38] (see Table A1). These results are well in accordance with the tested results examined based on the technical requirements in Vietnam, and they can be rephrased to show that the maximum mixing of RCB to RCA should be maintained at $f \leq 30\%$ when applying blends of RCA and RCB to unbound road base and subbase materials. The allowable limit of RCB mixed with RCA, which is $f \leq 30\%$ (on the safe side) in this study, moreover, matches well with the allowable limits suggested in previous studies [14,15,17].

In addition, the CBR value measured at K = 98% in this study was compared to the data on RCAs mixed with RCBs reported in the previous literature (Figure 4b). It is likely that the CBR values depend on $D_{max}$, and samples with lower $D_{max}$ generally gave higher CBR values. Most of the values with $D_{max}$ = 20 mm reported by Arulrajah et al. [32] exceeded 100% ($f$ < 50%) and became similar to the data of $D_{max}$ = 19 mm in this study. The CBR values with $D_{max}$ = 37.5 mm from Arisha et al. [49] exceeded 100% and became similar to the data of $D_{max}$ = 37.5 mm (well–graded) in this study within $30 \leq f \leq 60\%$. The CBR values with $D_{max}$ = 40 mm from Poon and Chan [14], however, did not exceed 100% ($f$ < 50%) and became lower than the data of $D_{max}$ = 37.5 mm (poor–graded) in this study.

### 3.3. Hydraulic Properties

Hydraulic properties are necessary for understanding the water balance and drainage in road base layers. Among the hydraulic properties, the saturated hydraulic conductivity ($K_s$) is the most important parameter in estimating the movement of water in the layers. The $K_s$ values of the tested samples when compacted at three different moisture conditions were measured as functions of $f$ and total porosity ($\phi$) and are shown in Figure 5. It was noted that the tested samples were compacted at three different initial moisture conditions—dry ($w_i$ = ~0.5–3%), $w_i$ = ~8% (close to MDD), and wet ($w_i$ = ~11–16%)—and the $\phi$ values were calculated by using $\phi = 1 - [G_s/(\text{packed dry density})]$. The $K_s$ values of $D_{max}$ = 37.5 mm (poor–graded) obtained were the highest among the tested samples and ranged in the order of $10^{-2}$ cm/s, irrespective of the differences in $f$ and $\phi$. This can be understood to indicate that the $K_s$ values of $D_{max}$ = 37.5 mm (poor–graded) were mainly controlled by the pathway of connected macropores formed in the tested samples, and the formation of connected macropores was not affected by the aggregate materials (i.e., the RCA and RCB) and $w_i$ in the compaction process. The $K_s$ values of the other tested samples, on the other hand, varied depending on $D_{max}$, $f$, and $w_i$. For the tested samples compacted in a dry condition (Figure 5a,b), the $K_s$ values gradually increased with increasing $f$, along with the increase in $\phi$, and reached the order of $10^{-4}$ cm/s. The $K_s$ when compacted at $w_i$ = ~8% (close to MDD) (Figure 5c,d) gave an almost constant values of ~$10^{-3}$ cm/s irrespective of $f$ and $\phi$. The $K_s$ compacted in the wet condition (Figure 5e,f) varied greatly depending on $f$, and those values increased from the order of $10^{-6}$ to ~$10^{-3}$ cm/s. In particular, a significant jump in $K_s$ after $f$ = 30% can be observed in accordance with the increase in $\phi$. The evidence observed from the results for $K_s$ in this study, therefore, suggests that many factors, including the $D_{max}$ and gradation (PSD) of the aggregates, mixing proportions of the RCA and RCB, and initial moisture condition in the compaction process, greatly affect the formation of water pathways and pore network structures, implying that further studies are needed to characterize pore network structures (e.g., [51,52]) in order to understand the hydraulic properties of unbound base and subbase materials.

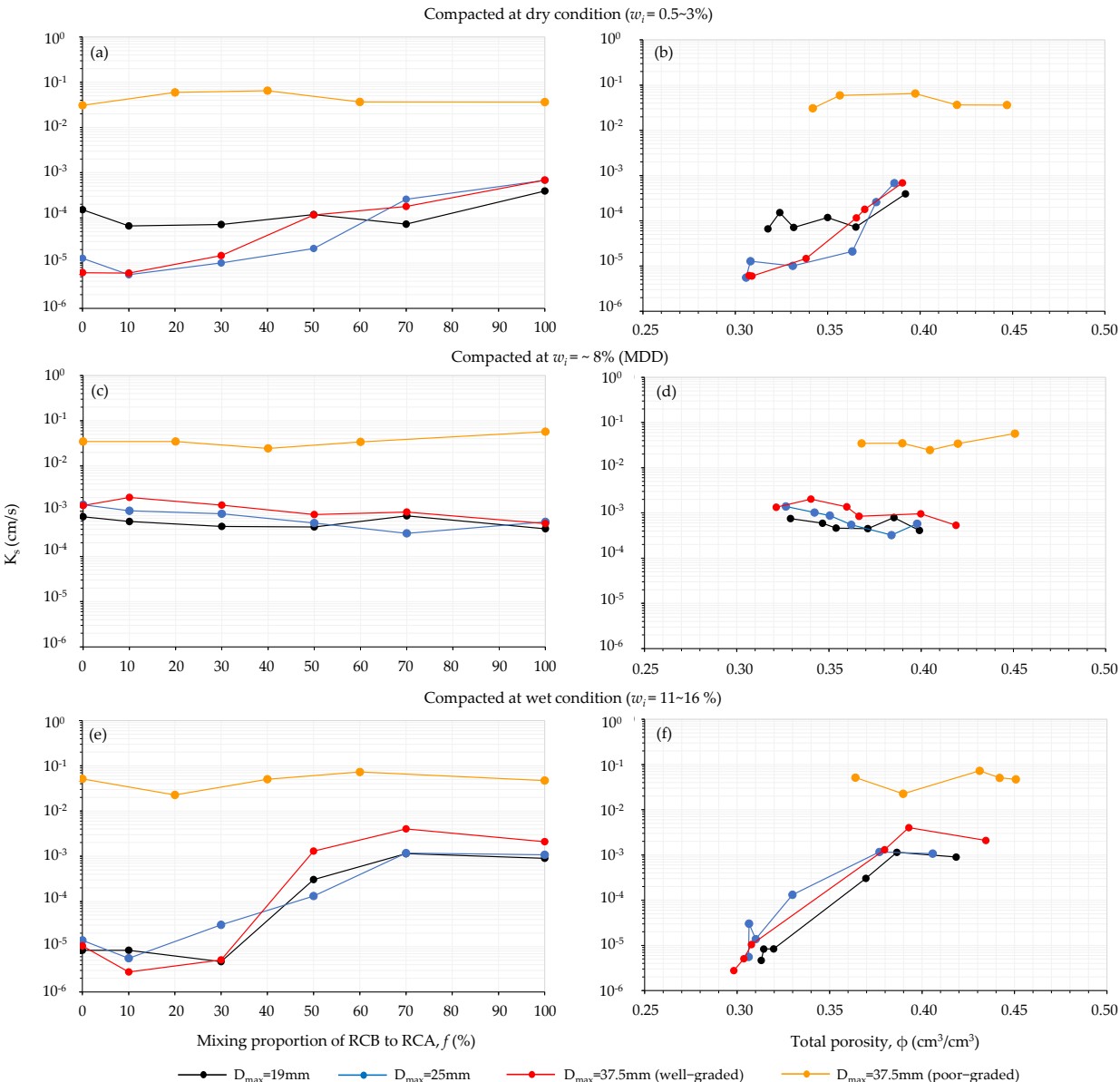

**Figure 5.** $K_s$ values of compacted samples in three different water conditions: dry ($w_i$ = 0.5–3%), $w_i$ = ~8% (MDD), and wet ($w_i$ = 11–16%). (**a**,**c**,**e**) $K_s$ as a function of mixing proportion ($f$ in %) of RCB to RCA. (**b**,**d**,**f**) $K_s$ as a function of total porosity, $\phi$ (cm³/cm³).

### 3.4. Particle Breakage Characteristics for Compacted Samples

3.4.1. Marsal's Breakage Index

The values of Marsal's breakage index ($B_g$) of the tested compacted samples, which were calculated with Equation (1), are summarized in Table 2 and shown as a function of $f$ in Figure 6. For all tested samples, the $B_g$ values generally increased with increasing $f$, indicating that samples with a high percentage of RCB became more breakable in the compaction process. The $B_g$ values measured at $f$ = 0% (RC 100%) and $f$ = 100% (RCB 100%) were close to previously reported values [33,35]. The $B_g$ values of $D_{max}$ = 19 mm gave smaller and less breakable values than those of the other tested samples, especially in the range of $f$ > 50%. This can be explained by the finding that the high amount of RCB with low $D_{max}$ prevented the breakage of aggregates in the compaction process (i.e., a cushioning effect [33,35]). It is interesting that the $B_g$ values of $D_{max}$ = 37.5 mm (well–graded) and $D_{max}$ = 37.5 mm (poor–graded) dissociated in the range of 10 < $f$ ≤ 50%; however, those values became closer and increased with increasing $f$ in the range of $f$ > 50%.

This could have been induced by the difference in the rearrangement of aggregates in the compaction process between the samples with different aggregate gradations, i.e., the greater rearrangement of aggregates for the poor–graded aggregates reduced (softened) the overall particle breakage compared to the well–graded aggregates in the compaction process in conditions of low RCB.

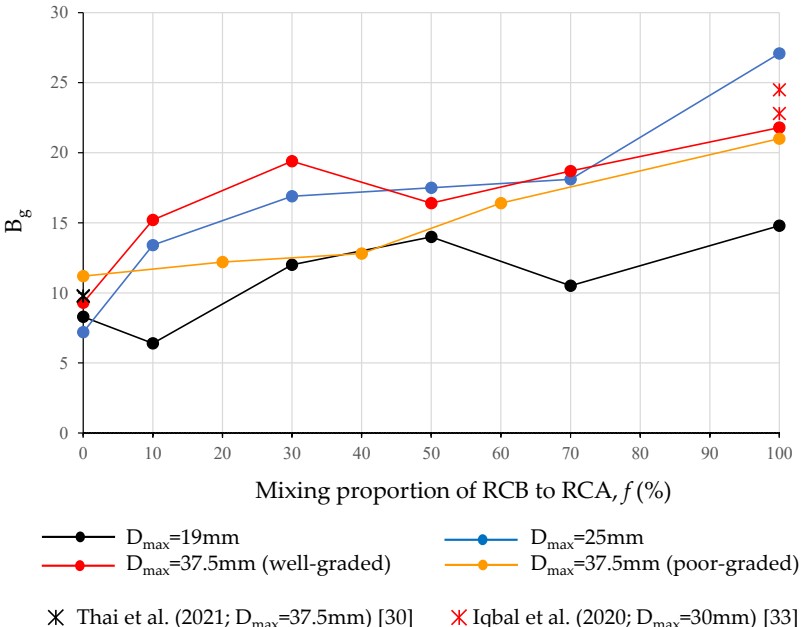

**Figure 6.** Calculated $B_g$ values as a function of the mixing proportion ($f$ in %) of RCB to RCA.

### 3.4.2. Particle Breakage at Each Size Fraction in the Compaction Process

Because the $B_g$ values represent the overall breakage characteristics based on the PSDs before and after compaction, they cannot give information on the particle breakage at each size fraction. Thus, to examine the details of the particle breakage characteristics, the percent increment and/or decrement in each particle fraction of the tested samples in three different initial moisture conditions—dry ($w_i$ = ~0.5–3%), $w_i$ = ~8% (close to MDD), and wet ($w_i$ = 11~16%)—was calculated and is shown in Figure 7.

For the tested samples with $D_{max}$ = 19 mm, the measured $B_g$ became smaller than those of the other samples (Figure 6), but it could be found that the percent decrement in 9.5–19 mm particles was relatively large and ranged from −3% to −8% for RCA 100% and from −12% to −16% for RCB 100% (Figure 7a–c). The percent decrement in 9.5–19 mm particles resulted in a percent increment (mostly <5%) in the particles smaller than 9.5 mm. For the tested samples with $D_{max}$ = 25 mm, the largest values of a percent decrement of −7 to −15% were observed in the 25–37.5 mm fraction for all tested samples (Figure 7d–f). For the percent increment, on the other hand, it could be found that the dependence on $w_i$ and the compacted samples at dry and $w_i$ = ~8% (close to MDD) water conditions gave high percent increments, with >15% observed for the 0.425–2.36 mm fraction for RCB 100% (Figure 7a,e).

For the tested samples with $D_{max}$ = 37.5 mm, a big difference was observed in the graphs of percent increment/decrement between the well–graded (Figure 7g–i) and poor–graded (Figure 7j–l) samples. For the well–graded samples, high percent decrements were found for the 19–25 and 25–37.5 mm fractions, and high percent increments were found with >5%, except for RCA 100%. In the poor–graded samples, on the other hand, the percent decrements in the coarse fractions of >19 mm ranged mostly in <5%, and a relatively high percent increment was observed in the 9.5–19 mm fraction. As shown in Figure 6, the $B_g$ values for the tested samples with $D_{max}$ = 37.5 mm that were well–graded and poor–graded became similar, especially in the range of $f$ > 50%. This evidence suggests that the

percent increment/decrement is effective in understanding the breakage characteristics of aggregates in the compaction process.

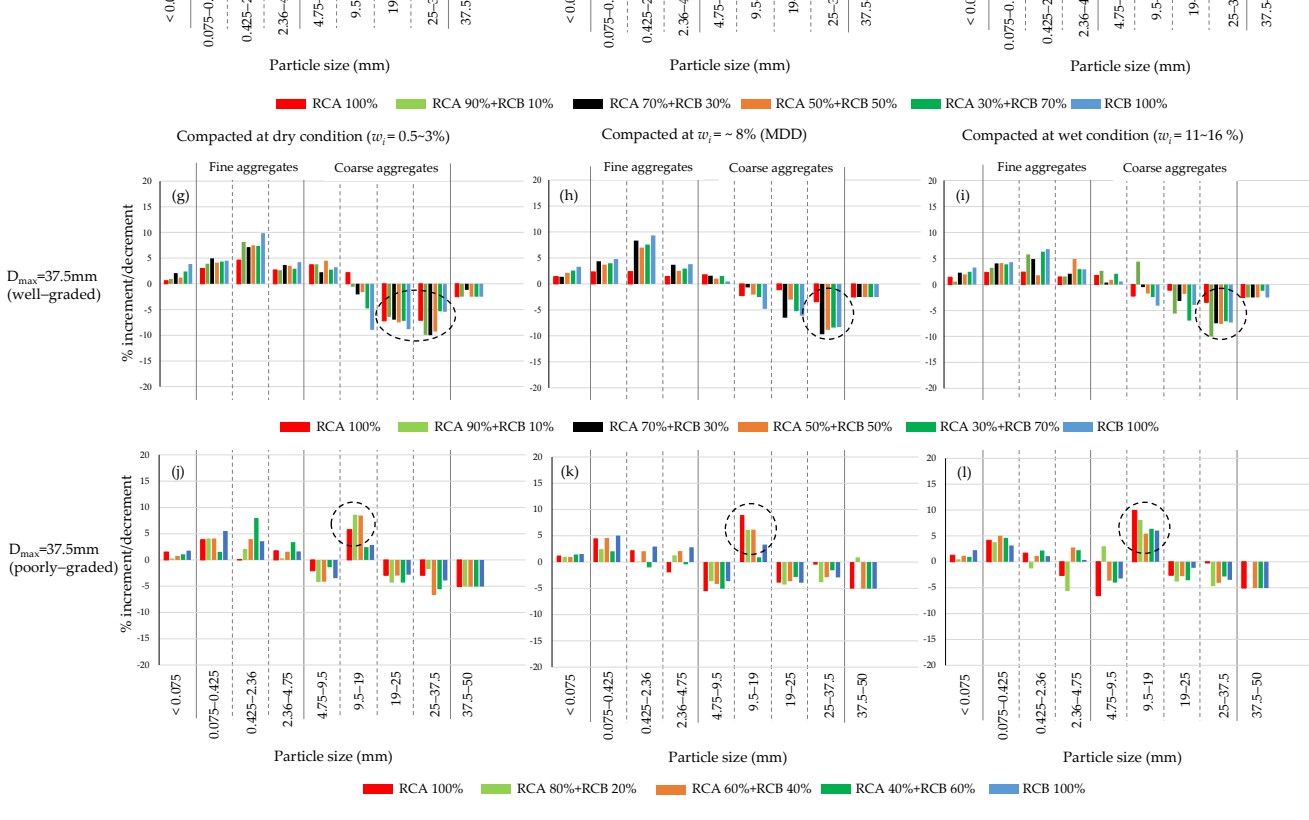

**Figure 7.** Percent increment/decrement of each retained fraction (dry-mass basis) before and after compaction of the tested samples with three different water conditions. (**a**–**c**) $D_{max}$ = 19 mm, (**d**–**f**) $D_{max}$ = 25 mm, (**g**–**i**) $D_{max}$ = 37.5 mm (well–graded), and (**j**–**l**) $D_{max}$ = 37.5 mm (poor–graded).

### 3.4.3. Percentages of RCA and RCB Retained at Each Fraction after Compaction

Finally, the percentages of RCA and RCB retained at each fraction for compacted samples with $D_{max}$ = 25 mm with $w_i$ = ~8% (close to MDD) were measured by using a combination of hand sieving and indirect estimation. The retained fractions with ≥2.36 mm were separable into concrete aggregates (RCAs) and clay brick aggregates (RCBs) by eye (i.e., hand sieving), as shown in Figure 8. The mixtures with fractions that were <2.36 mm, on the other hand, were difficult to separate by hand sieving. In this study, therefore, a new method of estimating the percentages of RCA and RCB is proposed considering the

properties of concrete and clay bricks: (i) Cementitious materials, including concrete, have a high mass loss due to being rich in hydrating water (thermal decomposition) and rich in Ca, and (ii) clay brick is low in mass and thermal decomposition due to its manufacturing process (burning) and is rich in Si and Al due to its origin from soil. The mass loss due to thermal decomposition (TG in %) was measured with a thermogravimetry/differential thermal analyzer (TG/DTA; TG/DTA6200, Hitachi High-Tech Corp., Tokyo, Japan). The elemental composition was measured through scanning electron microscopy with energy-dispersive X-ray spectroscopy (SEM-EDS; TM4000Plus, Hitachi High-Tech Corp., Tokyo, Japan, and AZtecOneGO, Oxford Instruments, Abington, UK). The measured atomic percentages of Ca, Si, and Al from the EDS analysis were used to determine the ratio of Ca/(Si + Al). Examples of the SEM and EDS images used to determine Ca/(Si + Al) are shown in Figure 9. It is noted that the tested samples were pre-heated at 400 °C for one hour to remove organic residues before the TG/DTA and SEM-EDS analyses.

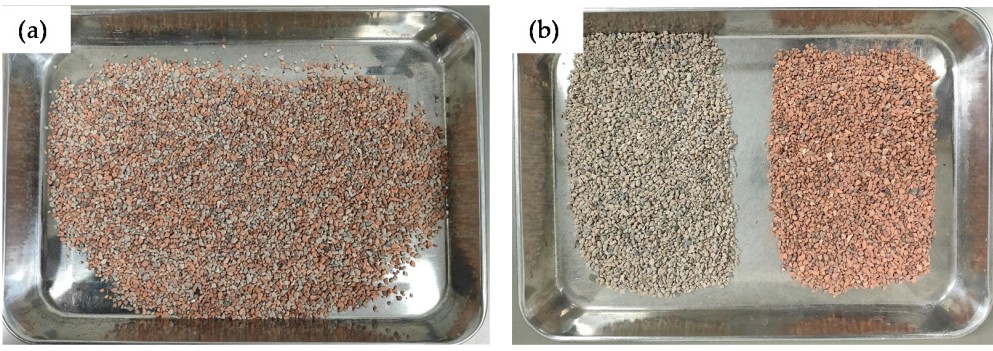

**Figure 8.** Hand-separated sample ($D_{max}$ = 25 mm; RCA 50% + RCB 50%; compacted at $w_i$ = 8%; MDD). The sample retained a fraction of 0.425–2.36 mm; (**a**) before separation and (**b**) after separation.

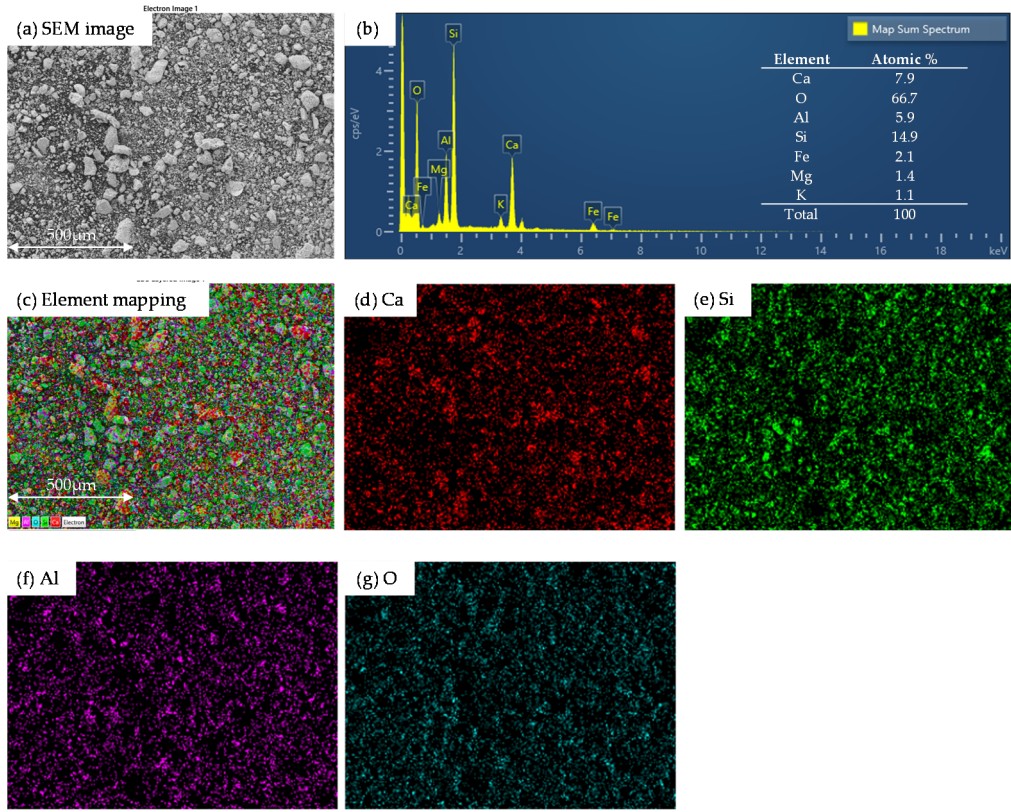

**Figure 9.** SEM-EDS images of a tested sample ($D_{max}$ = 25 mm; RCA 50% + RCB 50%; compacted at $w_i$ = 8%; MDD). The sample fraction is <0.075 mm.

In order to estimate the unknown percentages of RCA and RCB retained in the fine fractions of tested samples at <0.075, 0.075–0.425, and 0.425–2.36 mm, independent samples of RCA and RCB grains that were <2 mm were first mixed in three proportions: RC100%, RC 50% + RCB 50%, and RCB 100% (N = 10 for each sample). The measured TG (in %), Ca/(Si + Al), and their relationship are shown in Figure 10. As mentioned above, high TG and Ca/(Si + Al) for RCA 100% and low TG and Ca/(Si + Al) for RCB 100% were found, and a good linear relationship was obtained (Figure 10c). Then, a multi-regression analysis was carried out to correlate the mixing proportion ($f$ in %) of the measured TG and Ca/(Si + Al), and the multiple regression equation can be given as follows:

$$f = -8.7 \times TG + 29.2 \, \{Ca/(Si + Al)\} + 115 \ (r^2 = 0.99), \tag{3}$$

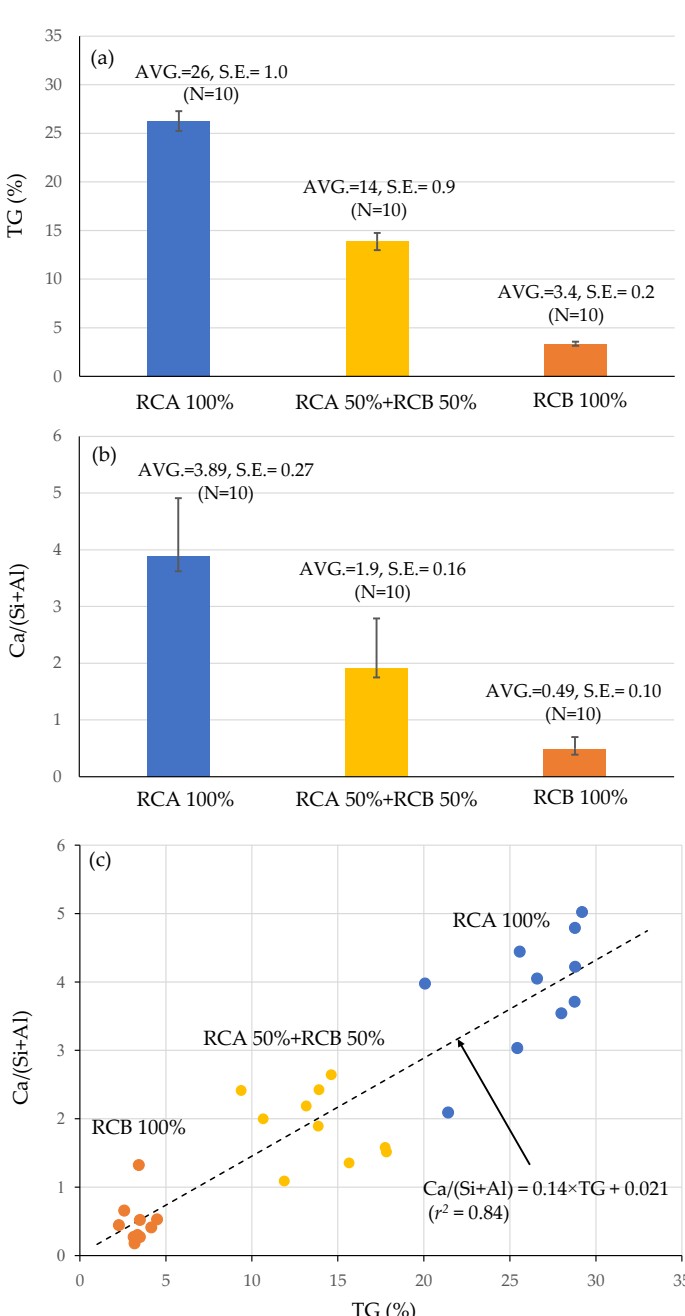

**Figure 10.** Measured (**a**) TG (%) and (**b**) Ca/(Si + Al) values for RCA 100%, RCB 100%, and RCA 50% + RCB 50% with a fraction of <2 mm (sample number, N is 10). Values of the average (AVG) and standard error (S.E.) are given. (**c**) Relationship between Ca/(Si + Al) and TG.

Thus, Equation (3) was used to estimate *f* (i.e., the percentages of RCA and RCB) of the unknown samples retained in the fine fractions.

The test results of the percentages of RCA and RCB retained at each fraction of the compacted samples (dry-mass basis) measured by using the proposed combination method are summarized in Table 3. In the table, not only the mass percentage of RCA and RCB, but also the mass-change percentage before and after compaction, is indicated using symbols, i.e., "↑" shows that the mass increment percentage is ≥10%, "↓" shows that the mass decrement percentage is ≥10%, and "→" shows that the mass change percentage is within 10%.

**Table 3.** Percentages of RCA and RCB retained at each fraction after compaction (dry-mass basis). "↑" shows that the change in the percentage of RCA/RCB after compaction increased by ≥10%. "↓" shows that the change in the percentage of RCA/RCB after compaction decreased by ≥10%. "→" shows that change in the percentage of RCA/RCB after compaction ranged within 10%.

| Sample: $D_{max}$ = 25 mm (Compacted at $w_i$ = ~8%: MDD) | | | | | | | | | |
|---|---|---|---|---|---|---|---|---|---|
| **Method** | | **Hand Sieving** | | | | | **Estimated by a Multi Regression Analysis** [1] | | |
| | **Fraction (mm)** | **25–37.5** | **19–25** | **9.5–19** | **4.75–9.5** | **2.36–4.75** | **0.425–2.36** | **0.075–0.425** | **<0.075** |
| RCA 90% + RCB 10% | RCA | 100 ↑ | 49 ↓ | 68 ↓ | 37 ↓ | 45 ↓ | 56 ↓ | 70 ↓ | 72 ↓ |
| | RCB | 0 ↓ | 51 ↑ | 32 ↑ | 63 ↑ | 55 ↑ | 44 ↑ | 30 ↑ | 28 ↑ |
| RCA 70% + RCB 30% | RCA | 100 ↑ | 82 ↑ | 74 → | 74 → | 71 → | 24 ↓ | 28 ↓ | 10 ↓ |
| | RCB | 0 ↓ | 18 ↓ | 26 → | 26 → | 29 → | 76 ↑ | 72 ↑ | 90 ↑ |
| RCA 50% + RCB 50% | RCA | 51 → | 48 → | 54 → | 50 → | 46 → | 23 ↓ | 26 ↓ | 23 ↓ |
| | RCB | 49 → | 52 → | 46 → | 50 → | 54 → | 77 ↑ | 74 ↑ | 77 ↑ |
| RCA 30% + RCB 70% | RCA | 49 ↑ | 35 → | 37 → | 37 → | 40 ↑ | 18 ↓ | 12 ↓ | 28 → |
| | RCB | 51 ↓ | 65 → | 63 → | 63 → | 60 ↓ | 82 ↑ | 88 ↑ | 72 → |

[1] $f = -8.7 \times TG + 29.2 \times \{Ca/(Si + Al)\} + 115$ (Equation (3); $f$: mixing proportion of RCB to RCA in %; TG in %).

For the tested samples with high-RCA mixtures (RCA 90% + RCB 10%, RCA 70% + RCB 30%), it was found that coarse RCB of 25–37.5 mm became zero (because all were crushed in the compaction process), and the percentage of RCB in fine fractions under 2.36 mm increased. This clearly indicated that the easily breakable coarse aggregates of clay brick were fully crushed to fines under the compaction process. For the tested sample of RCA 50% + RCB 50%, on the other hand, no significant percentage change was found for either the RCA or RCB with the fractions from 37.5 to 2.36 mm, and a high percent increment in RCB under 2.36 mm was observed (i.e., the original mixing proportion was maintained in the compaction process for fractions >2.36 mm). This indicates that the RCB fractions of >2.36 mm were uniformly crushed, resulting in an increment in fine fractions under 2.36 mm. For the tested sample with high RCB (RCA 30% + RCB 70%), the percent decrement in RCB mainly occurred in the two fractions of 25–37.5 and 2.36–4.75 mm, with a high percent increment in the two fractions of 0.425–2.36 and 0.075–0.425 mm. The test data in this study are still limited, and further studies are needed; however, the proposed method of quantifying the percent increment/decrement in RCA and RCB in the compaction process would contribute to understanding the mechanism of particle breakage in mixtures of different breakable aggregates.

## 4. Conclusions

A series of compaction, CBR, and saturated hydraulic conductivity tests of well–graded mixtures of RCB and RCA with $D_{max}$ = 19, 25, and 37.5 mm and a poor–graded mixture of RCA and RCB with $D_{max}$ = 37.5 mm were carried out in the laboratory, in which the mixing proportion of RCB to RCA (*f* in %) ranged from 0 to 100%. The particle breakage characteristics for the compacted RCA mixed with RCB at different initial moisture contents were analyzed based on the percent increment and/or decrement before and after compaction in this study. The major findings and conclusions in this study are as follows.

The MDD values gradually decreased with increasing *f*; however, there was no clear peak in dry density through the adjusted $w_i$, indicating that no OMC existed, except for the

tested samples with $D_{max}$ = 37.5 mm. In addition, the measured MDD values were generally smaller than those in the literature, especially in the range of $f$ > 30%. In addition, the PSDs significantly affected the compaction of all samples of the RCA, RCB, and their blends.

The CBR results of the tested samples at K = 98 and 95% suggest that the maximum mixing proportion of RCB to RCA should be maintained at $f \leq 30\%$ when applying mixtures of RCA and RCB to unbound road base and subbase materials, especially in Vietnam. The hydraulic test results showed that the saturated hydraulic conductivity was dependent on the $D_{max}$ and gradation of aggregates, mixing proportions of the RCA and RCB, and initial moisture content in the compaction process.

The characteristics of particle breakage were carefully examined by measuring the percent increment and/or decrement before and after compaction. It was suggested that determining the percent increment/decrement is effective for understanding the mechanism of the breakage characteristics of aggregates in the compaction process. In addition, an equation that enabled the estimation of the mixing proportions of the RCA and RCB retained in the fine fraction (<2.36 mm) was proposed based on a multi-regression analysis using measured the TG and Ca/(Si + Al). The proposed equation would contribute to understanding and characterizing the mechanism of particle breakage in mixtures of different breakable aggregates.

**Author Contributions:** Conceptualization, H.G.N., H.S. and K.K.; methodology, H.G.N., H.S. and K.K.; formal analysis, A.M., T.L.N. and K.K.; investigation, resources, and data collection, A.M., T.L.N. and V.T.N.; writing—original draft preparation, V.T.N., T.L.N. and K.K.; writing—review and editing, V.T.N., T.L.N. and K.K.; project administration, H.G.N. and K.K.; funding acquisition, H.G.N. and K.K. All authors have read and agreed to the published version of the manuscript.

**Funding:** This research was supported by the JST–JICA Science and Technology Research Partnership for Sustainable Development Program (SATREPS) project (No. JPMJSA1701).

**Institutional Review Board Statement:** Not applicable.

**Informed Consent Statement:** Not applicable.

**Data Availability Statement:** The data presented in this study are available on request from the corresponding author. The data are not publicly available due to the information security conditions of the project.

**Acknowledgments:** This research was supported by JST-JICA Science and Technology Research Partnership for Sustainable Development (SATREPS) (No. JPMJSA1701).

**Conflicts of Interest:** The authors declare no conflict of interest. The funders had no role in the design of the study; in the collection, analyses, or interpretation of data; in the writing of the manuscript; or in the decision to publish the results.

## Appendix A

**Table A1.** Materials and technical specifications of aggregates for road base and subbase layers in Vietnam and comparisons to those in Japan.

| | TCVN 8859: 2011 [36] | | | TCVN 8857: 2011 [37] | | | | JIS A 5001:1995 [53] | JRA: 2010 [38] | |
|---|---|---|---|---|---|---|---|---|---|---|
| Country | Vietnam | | | | | | | Japan | | |
| **1. Materials** | | | | | | | | | | |
| Labels | Type I | Type II | | Type A | Type B | Type C | Type D | C-40 | RC-40 | RM-40 |
| Materials | Crushed stone (natural aggregates) | (1) Crushed stone or gravel (100%) Crushed stone mixed with non-crushed natural aggregates (<50% content) | | Natural aggregates (minimized crushing processing) | | | | Crushed stone (natural aggregates) | Recycled aggregates (recycled materials from waste concrete and other materials) | Recycled aggregates (recycled materials from waste concrete and other materials) |
| **2. Gradation of aggregates** | | | | | | | | | | |
| Nominal aperture size of sieve (mm) | $D_{max}$ = 37.5 mm (1) (% passing) | $D_{max}$ = 25 mm (2) (% passing) | $D_{max}$ = 19 mm (3) (% passing) | $D_{max} \leq$ 50 mm (% passing) | | $D_{max} \leq$ 25 mm (% passing) | | | $D_{max}$ = 40 mm (% passing) | |
| 53 | | | | | | | | 100 | 100 | 100 |
| 50 | | | | 100 | 100 | | | - | - | - |
| 37.5 | 95–100 | 100 | | - | - | | | 95–100 | 95–100 | 95–100 |
| 26.5 | - | - | | - | - | | | - | - | - |
| 25 | - | 79–90 | 100 | - | 75–95 | 100 | 100 | - | - | 60–90 |
| 19 | 58–78 | 67–83 | 90–100 | - | - | | | 50–80 | 50–80 | - |
| 9.5 | 39–59 | 49–64 | 58–73 | 30–65 | 40–75 | 50–85 | 60–100 | - | - | - |
| 4.75 | 24–39 | 34–54 | 39–59 | 25–55 | 30–60 | 35–65 | 50–85 | 15–40 | 15–40 | 30–65 |
| 2.36 | 15–30 | 25–40 | 30–45 | - | - | - | - | 5–25 | 5–25 | 20–50 |
| 2.0 | - | - | - | 15–40 | 20–45 | 25–50 | 40–70 | - | - | - |
| 0.425 | 7–19 | 12–24 | 13–27 | 8–20 | 15–30 | 15–30 | 25–45 | - | - | 10–30 |
| 0.075 | 2–12 | 2–12 | 2–12 | 2–8 | 5–20 | 5–15 | 5–20 | - | - | 2–10 |

| | TCVN 8859: 2011 [36] | | | TCVN 8857: 2011 [37] | | | | JIS A 5001:1995 [53] | JRA: 2010 [38] | | |
|---|---|---|---|---|---|---|---|---|---|---|---|
| Country | Vietnam | | | | | | | Japan | | | |
| Labels | - | | | | | | | C-40 | RC-40 | RM-40 | |
| **3. Adaptations and technical specifications** | | | | | | | | | | | |
| Adaptations (5) | Base layer (A1, A2) | | Subbase layer (A1, Base layer (A2, B2) | (1) Subbase layer (A1: Type A, B, C) | (2) Base layer (A2: Type A, B, C) | (3) Subbase layer (A2: Type A, B, C, D) | (4) Base and subbase layers (B1, B2: Type A, B, C, D) | (5) Surface layer (B1, B2: Type A, B, C, D) | Testing method | Subbase layer (lower base) | Subbase layer (lower base) | Base layer (upper base) |

Testing method

**Table A1.** *Cont.*

| | | | | | | | | | | | | |
|---|---|---|---|---|---|---|---|---|---|---|---|---|
| LL (%) | ≤25 | ≤35 | ≤35 | ≤25 | ≤35 | ≤35 | ≤35 | [54] | - | - | - | - |
| PI | ≤6 | ≤6 | ≤6 | ≤6 | ≤6 | ≤12 | 9–12 | [54] | Non-plastic | ≤6 | ≤4 | [55] |
| PP index [6] | ≤45 | ≤60 | - | - | - | - | - | - | - | - | - | - |
| CBR at K = 98% (%) | ≥100 | - | ≥30 | ≥80 | ≥30 | ≥30 | ≥30 | [41] | - | - | - | - |
| Corrected CBR (%) [7] | - | - | - | - | - | - | - | - | - | ≥20 (30) [4] | ≥80 (90) [4] | [56] |
| LA (%) | ≤35 | ≤40 | ≤35 | ≤35 | ≤50 | ≤50 | ≤50 | [57] | ≤40 | ≤50 | ≤50 | [58] |
| Rate of sieve passing [8] | - | - | ≤0.67 | ≤0.67 | ≤0.67 | - | ≤0.67 | [59] | - | - | - | - |
| Elongation and flakiness index (%) | ≤18 | ≤20 | - | - | - | - | - | [60] | - | - | - | - |
| K (%) | ≥98 | ≥98 | - | - | - | - | - | [39] | - | - | - | - |
| Impurities (%) | - | - | - | - | - | - | - | - | ≤3 | ≤3 | ≤3 | [61] |
| Environmental safety | - | - | - | - | - | - | - | - | - | Satisfy environmental standards [9] | | MOE [10] |

LL: Liquid limit, PI: plastic index, CBR: California bearing ratio, LA: Los Angeles abrasion, K: compaction index (=degree of compaction). [1] Recommended for subbase: Type I and II; [2] recommended for base: Type I and II; [3] recommended for warping compensation and strengthening in old pavement structures when upgrading and renovating. [4] Values in parentheses: in case of use of recycled asphalt–concrete aggregates. [5] A1: Surface layer is made from hot-mix asphalt of class I [62]. A2: Surface layer is made from hot-mix asphalt of class II [62]. B1: Surface layer is made from crushed stone (macadam) with sand placing. B2: Surface layer is made from improved soil, local soil, and industrial refuse stabilized with a binder. [6] PP index = PI × (passing % of 0.075 mm); [7] CBR at K = 95%. [8] Rate of sieve passing = [(% of passing of 0.075 mm)/(% passing of 0.425 mm)]. [9] Satisfies environmental standards of Soil Contamination Countermeasure Law [63]. [10] Ministry of Environment (MOE) notification No. 46 [64].

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
