# Peer review of "Mechanical and Hydraulic Properties of Recycled Concrete Aggregates Mixed with Clay Brick Aggregates and Particle Breakage Characteristics for Unbound Road Base and Subbase Materials in Vietnam"

_sustainability, doi:10.3390/su14084854_

Round 1

Reviewer 1 Report

The manuscript presents an experimental study on the Recycled Concrete Aggregates Mixed with Clay Brick Aggregates in the application for Unbound Road Base and Subbase Materials. Several experimental results have been obtained and discussed. I have the following comment for the authors to improve their manuscript:

  • Fig. 3: the form of the curves does not clearly show the peak (optimum water content), please provide more analyses and discussion about the reasons of this particular point.
  • Fig. 4a: the values of MDD presented are relatively low for concretes (1.6-2.1 t/m3). Please check or provide the additional analyses/discussions.
  • Line 334: please provide some information about the saturated hydraulic conductivity, what is the objective to measure this parameter.
  • L. 459: "it was found that coarse RCB aggregates of 25–37.5 mm became zero" please revise this sentence to be clearer.
  • L 463-464: "no significant % change 463
    for both RCA and RCB aggregates with the fractions from 37.5 to 2.36 mm and high % increment of RCB under 2.36 mm" please revise to be clearer.

Reviewer 2 Report

A very good article. I have no comments.

Research should be continued. It would be good to know freeze-thaw resistance of mixtures.

Reviewer 3 Report

The paper proposed for publication presents experimental results on some properties characterizing recycled concrete aggregates in Vietnam. The paper can fit the Special Issue "Environmentally Sound Waste Management and Zero Waste Principles" since it covers the recycling of building waste considering regional characteristics.

The paper is well written, materials and methods are properly described but prior to publication I propose the following improvements:

  • Authors should better argument why this study is important for the scientific community. Authors made several experimental analysis but in the conclusion they are not recommending or envisaging any amendments to the Vietnamese’s standards to boost the use of such reclaimed material based on their findings.
  • Since the target application is Unbound Road Base and Subbase, I recommend, if possible to add a leachate testing to assess the environmental safety of this recycled material vs. its potential use.
  • For each of the property tested, section 2.2 and 2.3, authors should add a short indication about the values considered optimal or preferred for this or other (potential) applications, and why, in order to provide a reference or comparison.
